# Overcoming Barriers to Preventing and Treating *P. aeruginosa* Infections Using AAV Vectored Immunoprophylaxis

**DOI:** 10.3390/biomedicines10123162

**Published:** 2022-12-07

**Authors:** Jordyn A. Lopes, Amira D. Rghei, Brad Thompson, Leonardo Susta, Cezar M. Khursigara, Sarah K. Wootton

**Affiliations:** 1Department of Pathobiology, Ontario Veterinary College, University of Guelph, Guelph, ON N1G 2W1, Canada; 2Avamab Pharma Inc., 120, 4838 Richard Road SW, Calgary, AB T3E 6L1, Canada; 3Department of Molecular and Cellular Biology, University of Guelph, Guelph, ON N1G 2W1, Canada

**Keywords:** adeno-associated virus (AAV), monoclonal antibodies, *Pseudomonas aeruginosa*, nosocomial infections, chronic infections, cystic fibrosis, vectored immunoprophylaxis (VIP)

## Abstract

*Pseudomonas aeruginosa* is a bacterial pathogen of global concern and is responsible for 10–15% of nosocomial infections worldwide. This opportunistic bacterial pathogen is known to cause serious complications in immunocompromised patients and is notably the leading cause of morbidity and mortality in patients suffering from cystic fibrosis. Currently, the only line of defense against *P. aeruginosa* infections is antibiotic treatment. Due to the acquired and adaptive resistance mechanisms of this pathogen, the prevalence of multidrug resistant *P. aeruginosa* strains has increased, presenting a major problem in healthcare settings. To date, there are no approved licensed vaccines to protect against *P. aeruginosa* infections, prompting the urgent need alternative treatment options. An alternative to traditional vaccines is vectored immunoprophylaxis (VIP), which utilizes a safe and effective adeno-associated virus (AAV) gene therapy vector to produce sustained levels of therapeutic monoclonal antibodies (mAbs) in vivo from a single intramuscular injection. In this review, we will provide an overview of *P. aeruginosa* biology and key mechanisms of pathogenesis, discuss current and emerging treatment strategies for *P. aeruginosa* infections and highlight AAV-VIP as a promising novel therapeutic platform.

## 1. Introduction

*Pseudomonas aeruginosa* (*P. aeruginosa*) is recognized as a pathogen of global importance and is among the list of ESKAPE pathogens (this includes *Enterococcus faecium*, *Staphylococcus aureus*, *Klebsiella pneumoniae*, *Acinetobacter baumannii*, *Pseudomonas aeruginosa*, and *Enterobacter species*) that exhibit both multidrug resistance and virulence [1]. Taking into consideration the global burden of disease, patient morbidity and multidrug resistance, the World Health Organization has listed ESKAPE pathogens as a top priority for which effective antibiotics are urgently needed [2]. *P. aeruginosa* is a nosocomial bacterial pathogen that is particularly prevalent in patients suffering from cystic fibrosis (CF). Roughly 40% of patients with CF have chronic *P. aeruginosa* infections by the time they reach adulthood [3], which is associated with worsening lung function and increased mortality [4,5]. Antibiotic treatment is the first line of defense when *P. aeruginosa* is first detected, although successful eradication is variable and not sustained [6]. Since *P. aeruginosa* has acquired intrinsic resistance to multiple classes of antibiotics, and there are currently no licensed vaccines for the prevention and treatment of this pathogen, alternative treatment approaches, such as antibody-based strategies, to prevent *P. aeruginosa* infection and persistence are needed. In this review, we will discuss the history, pathogenesis, and current treatment strategies of *P. aeruginosa*, emphasize the connection between nosocomial infections and chronic infections in patients with CF, examine the application of antibody therapy for *P. aeruginosa* infections, and delve into the potential use of adeno-associated virus vector-mediated expression of monoclonal antibodies (mAbs) as an alternative prophylactic platform for long-term passive immunity against *P. aeruginosa* infections.

## 2. *P. aeruginosa*

### 2.1. History of Hospital Acquired P. aeruginosa Infection

*P. aeruginosa* was first observed in 1850 by Sédillot [7]; however, it was not until an article by Freeman in 1916 that routes of entry and dissemination in hosts, leading to either acute or chronic infection by *P. aeruginosa*, were described [8]. *P. aeruginosa* is a ubiquitous microbe that can survive in diverse environments, such as living plants, animals, and humans, due to its robust ability to thrive off minimal nutrients and tolerate harsh conditions. This vigorous survival has led this pathogen to persist in both community and hospital settings. *P. aeruginosa* has been isolated in communities from swimming pools, hot tubs, home-use humidifiers, soil, and vegetation, and has been found in respiratory therapy equipment, antiseptics, sinks, cleaning equipment, medicine and hydrotherapy pools [7,9,10]. Even with such a vast distribution of *P. aeruginosa* in nature and in communities, most serious infections with *P. aeruginosa* are hospital acquired.

A hospital acquired infection (HAI) is defined as an infection a patient develops while receiving medical care that is not associated with the patient’s original diagnosis. In recent years, the financial burden attributed to HAIs in healthcare settings has increased substantially. It is estimated that the overall direct cost of HAIs in the United States ranges from $28 billion to 45 billion USD, annually [11]. While the causative agents of HAIs can be attributed to viral, fungal or parasitic infections, bacterial infections are the most common cause of HAIs on a global scale [12]. Nosocomial infections caused by Gram-negative bacteria, inclusive of *P. aeruginosa,* are of particular concern due to the low permeability of the bacterial cell wall [13], and capacity to both acquire and upregulate intrinsic resistance mechanisms against antibiotic treatment [14].

*P. aeruginosa* is an opportunistic pathogen frequently associated with persistence in healthcare settings, given its presence in hospital water, healthcare equipment, and staying power on dry inanimate objects from 6 h to 6 months [13]. Surveillance data gathered by the Centers for Disease Control and Prevention National Nosocomial Infections Surveillance System (NNIS) between 1986 to 1998 reported *P. aeruginosa* as the fifth most isolated nosocomial pathogen and the cause for 9% of all nosocomial-acquired infection in the United States [15]. Furthermore, the NNIS reported 21% of pneumonias, 10% of UTIs 3% of bloodstream infections and 13% of ear, eye, nose and throat infections detected within the ICU in the United States were caused by *P. aeruginosa* [16]. Through extensive studies on the burden of *P. aeruginosa* on the healthcare system were heavily monitored up until the early 2000s, the majority of the *P. aeruginosa* reporting since has heavily described the impact of antimicrobial resistance (AMR), thus limiting our knowledge of the true impact of *P. aeruginosa* infections.

### 2.2. Taxonomy

Belonging to the *Gammaproteobacteria* class, and *Pseudomonadaceae* family, the *Pseudomonas* genus is among a class of Gram-negative bacteria containing 302 validly named species (List of Prokaryotic Names with Standing in Nomenclature (https://lpsn.dsmz.de/genus/pseudomonas, accessed on 13 October 2022). *Pseudomonas* is one of the largest and most complex Gram-negative bacteria with the first complete genome sequence of strain PA01 being reported in 2000 [17]. From March 2020 to March 2021, more than 30 new *Pseudomonas* species were reported, with continuous reporting of new species [18]. Though the taxonomy of bacterial species was previously organized by phenotypic and metabolic features alone, genomic tools, such as 16S rRNA gene sequencing, DNA-DNA hybridization and GC-content profiling, have enabled a polyphasic taxonomic approach [18].

### 2.3. Molecular Biology and Structural Characteristics of P. aeruginosa

*P. aeruginosa* is a Gram-negative, facultative aerobic, non-spore forming and bacillus-shaped bacterium, considered to be saprophytic in nature [19,20]. *P. aeruginosa* has a circular genome (~5–7 Mbp, strain-dependent) containing over 5000 core genes conserved amongst all isolates [21]. These core genes conserved among all species represents up to 90% of the bacterial genome, and consists primarily of housekeeping genes [22]. Accessory genomic elements of up to 200 kbp are moderately variable and are indicative of the ability of *P. aeruginosa* to acquire genetic elements through horizontal gene transfer (either transformation, conjugation or transduction) [22]. For example, two genomic islands that harbor pathogenic mechanisms specific to *P. aeruginosa* are *P. aeruginosa* pathogenicity island 1 (PAPI-1) and PAPI-2, which both encode numerous virulence genes and are well characterized in the highly virulent clinical strain PA14 [23]. Furthermore, accessory genome elements play a major role in the ability of *Pseudomonas* to persist in various environments, especially in clinical settings which undergo continuous sanitation [22,23]. Intrinsic antibiotic resistance mechanisms are encoded within the core genome, while acquired antibiotic resistance is obtained through accessory genomic elements [22]. The acquired antibiotic resistance is transferred amongst strains, which contributes to multi-strain outbreaks in healthcare settings.

Lipopolysaccharide (LPS) is a structural component of the cell envelope seen in Gram-negative bacteria and is considered a major virulence factor in *P. aeruginosa* pathogenesis. LPS consists of a lipid A domain, a core oligosaccharide and O-antigen, all of which play crucial roles in bacterial physiology and protection against the host defense system [24]. The main function of the LPS is to provide structural integrity to the outer membrane (OM), in addition to playing key roles in colonization and virulence. *P. aeruginosa* consists of a very similar structure to other Gram-negative bacteria, with a cell wall consisting of an inner membrane, peptidoglycan layer and outer membrane [25].

### 2.4. Mechanisms of Pathogenesis

The opportunistic nature of *P. aeruginosa* enables it to cause disease in a variety of organisms including plants, insects, and mammals. *P. aeruginosa* is of particular concern in hospital settings due to its capacity to colonize and persist on artificial surfaces, posing an extreme risk to elderly and immunocompromised individuals [26]. This opportunistic pathogen can be readily isolated from the skin, throat, and stool of healthy individuals, and transmission occurs through individuals via contact with fomites, ingestion of contaminated food, water or aerosols, or entry through wound infections [3]. Once *P. aeruginosa* enters the host, it deploys a diverse arsenal of mechanisms that can be exploited for successful infection and evasion of the host immune system. Many gene functions have been shown to contribute to *P. aeruginosa* pathogenesis, including those involved in biofilm formation, quorum sensing, protein secretion, LPS, outer membrane vesicle (OMV) formation, and bacterial motility [19].

#### 2.4.1. Lipopolysaccharide as a Virulence Factor

During infection, LPS induces toll-like receptor 4 (TLR-4) dependent and independent inflammatory responses, and also stimulates host immune responses by activating the complement system and neutrophil extracellular trap formation (NETosis), resulting in programmed cell death and inhibition of bacterial dissemination [27,28]. The O antigen is necessary for bacterial motility, including swimming and swarming, which are heavily implicated in bacterial pathogenesis [12,24]. These diverse motility properties are utilized to sense, search, and migrate bacterial cells towards the surface, thus enabling bacterial cell-surface contact, transitions quickly from sessile to a planktonic lifestyle, and transitions from reversible to irreversible attachment for biofilm formation to occur [12]. Studies have shown that the planktonic lifestyle of *P. aeruginosa* facilitates the rapid release of effector toxins resulting in an acute infection, while the sessile lifestyle is attributed to chronic infections through biofilm formation, capable of attaching to both biotic and abiotic surfaces. Both lifestyles are influenced by bacterial motility and are understood to play an important role in the sustained virulence of *P. aeruginosa* [12].

#### 2.4.2. Outer Membrane Vesicles as Virulence Factors

OMVs are bacterial components approximately 10 to 300 nm in diameter that function to deliver a variety of bacterial proteins, lipids, nucleic acids, ion metabolites, and signaling molecules in order to increase virulence capacity [28]. *P. aeruginosa* OMVs have been linked to the priming of host tissue surfaces for bacterial adhesion, acquisition of drug resistance, immune evasion, as well as the removal of competing bacteria from the environment during an infection [27,28]. Interestingly, OMVs also reduce cystic fibrosis transmembrane conductance regulator (CFTR) protein Cl¯ ion secretion in bronchial epithelial cells, thus reducing bacterial clearance from the lungs [28].

#### 2.4.3. Role of Biofilm Formation

Biofilm formation is advantageous to the survival and protection of bacteria in the face of host immune responses, antibiotics, as well as other adverse environmental conditions including fluctuations in pH, temperature and nutrient availability [19]. It is important to note that bacteria encased within a biofilm can resist exogenous stressors up to 1000 times more than free-living bacteria [19]. Consisting of approximately 90% of the biofilm biomass, the *P. aeruginosa* biofilm matrix is comprised of polysaccharides, extracellular DNA, lipids and proteins. Three exopolysaccharides (Pei, Psl and alginate) are produced by *P. aeruginosa* and play crucial roles in biofilm formation, surface attachment, biofilm stability, and bacterial clearance [26,27]. *P. aeruginosa* biofilm development is typically slow growing, as unattached cell aggregates thrive under hypoxic and anoxic conditions. These conditions are commonly seen in the airways of people with CF (pwCF), which may play a role in their heightened susceptibility to *P. aeruginosa* infections [26]. Studies have shown that sessile cells in the heterogenic biofilm communities show a marked decrease in metabolic activity throughout the stationary-like phase of the bacterial growth cycle. This decrease in metabolic activity is suggested to contribute to the increased tolerance of biofilms against antimicrobial agents in comparison to free-living cells [26,29].

#### 2.4.4. Mechanisms of Quorum Sensing

Quorum sensing is the regulation of gene expression in response to fluctuations of cell density within a population [30]. These communication circuits enable specific colony-wide functions and regulate processes such as symbiosis, virulence, conjugation, motility, sporulation and biofilm formation [30]. *P. aeruginosa* autoinducer signal molecules *N*-homoserine lactone and *N*-butyryl homoserine lactone accumulate as the bacterial population increases. Once the intracellular threshold is surpassed, these molecules bind and activate their transcription factors LasR and RhlR, which regulate hundreds of genes [31]. LasR and RhlR are known to facilitate the production of numerous virulence factors including elastase, alkaline protease, exotoxin A, rhamnolipids, and lectins [31,32].

#### 2.4.5. Secretion Systems and Their Integral Role in Virulence

To successfully invade the mammalian host cell, *P. aeruginosa* utilizes a complex set of secretion systems to transport virulence factors from the bacterial cell envelope across the host cell plasma membrane. These processes are critical for bacterial invasion of host tissue sites and manipulation of the host immune responses. Secretion systems function to secrete proteins across the inner and outer membrane of the cell envelope of the phospholipid membrane, contributing to the growth and survival of Gram-negative bacteria [33]. Exoproteins contribute to bacterial virulence by upregulating cell attachment, and directly impacting cellular function through intoxication via the excretion of enzymes, proteins, and toxins. Currently, the opportunistic *P. aeruginosa* pathogens possess six known secretion systems found in Gram-negative bacteria, sometimes with multiple copies per bacterium [27,33]. Secretion systems are broadly categorized as either a one-step system, which involves secreting proteins directly from the bacterial cytosol to the surface, and are independent of the secretory pathway, or a two-step system which results in the brief translocation of secreted proteins into the periplasm before being diffused into the extracellular space [34]. These systems are considered Sec-dependent, as they require either the Sec translocon or Tat pathway in order to transfer secreted proteins across the inner membrane (IM) [34]. *P. aeruginosa* possesses Type II secretion system (T2SS) and T5SS, which are classified as one-step secretion systems while the T1SS, T3SS, T4SS, and T6SS are classified as two-step secretion systems [27,34]. The diversity of secretion systems available to this opportunistic pathogen enables the production and release of a wide variety of exoproteins, thus leading to enhanced bacterial adaptation to the environment and/or the host [33].

Two protein secretion systems are highlighted as major contributors to virulence in *P. aeruginosa*, namely T2SS and T3SS. The T2SS is responsible for the secretion of the most toxic virulent factor, exotoxin A (ETA). This toxin is an ADP-ribosyl transferase, capable of decreasing amino acid uptake and inactivating the eukaryotic elongation factor 2 (eEF-2), thereby inhibiting host protein synthesis [35,36]. ETA also activates two caspases involved in apoptosis, inhibits IL-18 secretion, and has been shown to decrease the production of TNF-α, IL-6, IL-8, and IL-10 [36]. This toxin is extremely lethal to mammalian cells and has been shown to possess an LD50 of 0.2 µg per mouse via intraperitoneal injection [37].

The T3SS is highlighted as a contact-dependent protein secretion pathway and is heavily implicated in the pathogenesis of acute *P. aeruginosa* infections [38]. A needle-like injectosome is required to penetrate through the host cell wall to facilitate translocation of various effector proteins into the cytoplasm. The four main effectors are known as ExoS, ExoT, ExoU, and ExoY [33,39]. ExoS/ExoU induce host cell apoptosis and colonization through the targeting of the c-Jun NH2 terminal kinase (JNKs) signal pathway, while ExoY/ExoT inhibit bacterial motility to dampen the nuclear factor kappa B (NF-kB) and caspase-1 activation, thereby reducing inflammasome activity from the host. Attempts to directly target and inhibit effector proteins have not been successful to date, although targeting the physical protein which is localized at the apex of the injectosome, called *P. aeruginosa* V-antigen (PcrV), has therapeutic applications as it has been shown to be successful in blocking the release of these effector proteins [39].

## 3. *P. aeruginosa* and Cystic Fibrosis

The diverse repertoire of virulence factors utilized by *P. aeruginosa* has led to this pathogen as one of the most common Gram-negative bacteria causing nosocomial infections and is reported to be responsible for 10–15% of nosocomial infections worldwide [40]. Infection of immunocompetent individuals with *P. aeruginosa* can result in a variety of clinical manifestations, such as urinary tract infections, respiratory infections, dermatitis, soft tissue infections, bacteremia, bone and joint infections, and gastrointestinal infections [41]. These clinical manifestations are much more severe in immunocompromised individuals, often resulting in complications that can lead to death. Individuals with cancer, neutropenia, severe burns, and in particular pwCF are at a much greater risk of serious complications when infected with *P. aeruginosa* [41].

CF is the most common monogenic disorder in North America, caused by recessive mutations in the CFTR gene [42]. This gene is responsible for regulating the transport of chloride and electrolytes across the epithelial cell membrane, enabling normal cell homeostasis. The absence or dysfunction of this protein results in the build-up of thick mucus layers in the lung, hindering mucociliary clearance of potential pathogens, which can lead to a wide range of pathological consequences including serious chronic lung infections and inflammation [24]. CF patients are highly susceptible to opportunistic bacterial pathogens, specifically *P. aeruginosa*. A study conducted by Douglas and colleagues reported that 28% of children with CF acquired pulmonary infections with *P. aeruginosa* within the first 6 years of life [43]. Another study determined that the risk of adults with CF death doubled with *P. aeruginosa* present and increased by eightfold when *P. aeruginosa* acquired antibiotic resistance [44]. As *P. aeruginosa* is frequently the most dominant organism in CF airways during the end-stage of the disease [24], it is critical that *P. aeruginosa* infections are diagnosed and eradicated prior to the overproduction of polysaccharide alginate, which results in the transition from non-mucoid to mucoid phenotype. The mucoid phenotype is suggested to enhance biofilm formation and is a hallmark of chronic and persistent infections [24].

*P. aeruginosa* is an extremely successful pathogen due in part to its ability to adapt and proliferate in niche environments, which can be attributed to its intrinsic and acquired resistance capabilities [45]. Various antibiotic survival mechanisms are utilized by *P. aeruginosa* to support the survival of persister cells, an antibiotic tolerant phenotype of bacteria which serves to repopulate the bacterial population after the course of antibiotic treatment has been completed [45]. Classic antibiotic resistance can be conferred through mutations and/or horizonal gene transfer of antibiotic-modifying enzyme genes, which can be acquired through plasmid transfer from one bacterial species to another [45]. Adaptive resistance, described as the transient resistance to antibiotics, typically emerges rapidly in the response to prolonged exposure to antibiotic treatment. Antibiotic resistance is a significant problem in clinical settings, and without novel treatment approaches, will only progress. A study determined that out of 60 clinical isolates of *P. aeruginosa* strains from burn patients, 90% were resistant to at least one antibiotic, while 94% were considered multidrug resistant (MDR) [45].

## 4. Pathogen-Host Interactions in *P. aeruginosa* Infections

In order to effectively eradicate *P. aeruginosa* from the lungs, a combined effort of both host innate and adaptive immune recognition is required to clear infection. Invasion of *P. aeruginosa* triggers a robust immune response during the acute phase of the infection [46]. Although this is necessary for bacterial clearance, often an exacerbated inflammatory response can cause bacterial persistence and tissue damage, leading to poor host outcomes [46].

TLRs are understood to play an important role in the activation of innate immunity in *P. aeruginosa* infections. Upon host recognition, the LPS can activate TLR-4, followed by TLR-5 activation through *P. aeruginosa* flagella. TLR-9 is activated by unmethylated bacterial CpG DNA, which is suggested to contribute to the sensing of *P. aeruginosa* bacteria and extracellular-DNA-containing biofilms [47]. *P. aeruginosa* can also facilitate the release of OMVs. These OMVs can deliver peptidoglycan to nucleotide–binding oligomerization main containing protein-1 (NOD-1) receptors, which can then interact with the receptor interaction protein-2 (RIP-2) adaptor protein to activate NF-κB and mitogen-activated protein kinase pathways in epithelial cells [47]. The bacterial nucleoside diphosphate kinase of *P. aeruginosa* has been found to cause cytotoxicity in macrophages [47].

The T3SS plays a major contributor to *P. aeruginosa* virulence due to its ability to directly inject bacterial effectors into host cells and mediate bacterial recognition through PRRs, leading to the activation of inflammasomes. In addition to the T3SS effector proteins stimulating an innate immune response, the T3SS needle can activate the NLR Family Card Domain 4 (NLRC4) inflammasome through neuronal apoptosis inhibitory protein (NAIP) recognition, leading to pyroptotic cell death and the secretion of IL-1ß and IL-18 [48]. Furthermore, studies have shown that in patients with chronic *P. aeruginosa* infections, antibodies against T3SS-dependent effector proteins are present, although strains isolated from chronic infections are typically T3SS-negative. This suggests that CF-adapted *P. aeruginosa* can downregulate T3SS expression in order to enhance immune evasion and promote bacterial persistence, and further suggesting that the T3SS may not play as important a role in virulence during chronic infections as during acute infections [48].

Neutrophil recruitment is considered a vital initial innate immune response that is commonly seen in chronic *Pseudomonas* lung infections. One caveat to this effective host defense mechanism is that chronic stimulation of neutrophils by bacterial pathogen-associated molecular patterns (PAMPs) and/or cytokines can greatly increase local damage through granulation and inflammation [49]. Overstimulation of neutrophils often leads to neutrophil dysfunction and the inability for neutrophils to clear a bacterial infection. *P. aeruginosa* employs numerous mechanisms to evade and eliminate neutrophils, including bacterial adsorption of host sialo-glycoproteins to inhibit degranulation, reactive oxygen species (ROS) production and NETosis. Interestingly, studies have shown that in cases of chronic CF, *P. aeruginosa* is capable of not only developing resistance to neutrophil effector mechanisms, but also using neutrophils to promote biofilm growth in CF airways. Extracellular DNA (eDNA) obtained from necrotic neutrophils has been shown to support the growth of *P. aeruginosa* aggregates, which is required for biofilm formation [50].

In addition to host innate immunity, adaptive immunity is commonly seen in the presence of biofilm formation, and plays critical roles in *P. aeruginosa* pathogenesis [50]. In bacterial infections, the balance between T lymphocyte helper (T_h_)1 and T_h_2 cells are typically skewed, thereby impairing or completely inhibiting bacterial clearance [51]. A study conducted by Moser et al., showed that in chronic vs. non-chronically infected CF patients the levels of T_h_2 marker IL-4 were significantly higher in chronically infected patients, as opposed to non-chronically infected patients who showed elevated levels of IFN-γ, produced by macrophages and T_h_1 helper cells [52]. This suggests that biofilm formation seen in chronic *P. aeruginosa* infections can skew the immune response in favor of the pathogen.

Despite a wide array of host immune defense mechanisms that are understood to play a role in the clearance of *P. aeruginosa* infections, this opportunistic pathogen is capable of quickly adapting to its environment, which presents a major problem in nosocomial settings where previously effective treatment strategies are becoming less effective with the rise of antimicrobial resistance.

## 5. Current Treatment Strategies for *P. aeruginosa*

### 5.1. Antibiotic Treatment Strategies Currently Available

To date, the only approved treatment strategy for *P. aeruginosa* infections is antibiotic therapy. Known risk factors and the progression of infection both play vital roles in treatment plan development and should be evaluated on a case-by-case basis. Typically, if infection is severe, empirical antibiotic treatment will be started, which should include two different classes of antibiotics to increase the likelihood of effective treatment, especially with the rise in antimicrobial resistance. Typically, treatment relies on in vitro susceptibility testing, and should be tailored accordingly whenever possible [53]. Combination and then de-escalation to monotherapy are often recommended.

Aminoglycoside and beta-lactam penicillin is the first line of treatment for *P. aeruginosa* infections [54], although there are numerous broad spectrum antibiotics that have also shown to be effective against *P. aeruginosa* infections (Table 1) [55]. In addition to the variety of antibiotics used to treat *P. aeruginosa* infections, the combination of Β-lactam antibiotics such as penicillin and ß-lactamase inhibitor has been shown to have synergistic effects, increasing the effectiveness of ß-lactam to antibiotics [56].

### 5.2. Increasing Antimicrobial Resistance of P. aeruginosa

Antimicrobial resistance is a major threat to public health worldwide, with serious implications for the future of our current and most effective treatment against bacterial infections: antibiotics. A comprehensive global burden analysis estimated 4.95 million deaths associated with bacterial AMR alone in 2019, with 1.27 million deaths attributed to bacterial AMR [59].

Six bacterial pathogens possessing multidrug resistance and extreme virulence have been termed ‘ESKAPE’ pathogens—an acronym for *Enterococcus faecium*, *Staphylococcus aureus*, *Klebsiella pneumonia*, *Pseudomonas aeruginosa*, and *Enterobacter* spp. [60]. These pathogens are listed as priority pathogens by the World Health Organization (WHO) due to their resistance to multiple classes of antibiotics and pose the greatest threat to human health. Carbapenem-resistant *P. aeruginosa* is among the group listed as priority 1, due to its ability to cause severe infection in nosocomial settings, which can lead to deadly infections including bacteremia and pneumonia [60]. Recently, *P. aeruginosa* has been listed as one of the six leading pathogens responsible for causes of AMR deaths, which have been termed ‘SPEAKS’—an acronym for *Staphylococcus aureus*, *Pseudomonas aeruginosa*, *Escherichia coli*, *Acinetobacter baumannii*, *Klebsiella pneumonia* and *Streptococcous pneumoniae* [59,61]. Various antibiotic survival mechanisms are utilized by *P. aeruginosa* to support the survival of persister cells, an antibiotic tolerant phenotype of bacteria, which serve to repopulate the bacterial population after the course of antibiotic treatment has been completed. Studies have shown that these bacterial persister cells can form in the biofilm and are responsible for recurrent infections seen in CF patients [62]. *P. aeruginosa* possesses 12 resistance–nodulation–division (RND) type efflux pumps that play crucial roles in the reduction of antibiotic susceptibility. These MDR efflux pumps have also been shown to play a role in the acquisition of MDR [42]. Studies have shown that constitutive overexpression of MDR efflux pumps facilitated through the deletion of regulatory genes increases bacterial resistance [63].

### 5.3. Antimicrobial Resistance of P. aeruginosa in a Clinical Context

Antimicrobial resistant strains of *P. aeruginosa* are quickly becoming a major concern in clinical settings, especially due to the increasing development of multidrug resistant (MDR) strains. Intrinsic resistance to antimicrobials including B-lactam and –penem group of antibiotics is attributed to the low permeability of its outer membrane. Upregulation of efflux pump systems and intrinsic production of antibiotic inactivating enzymes are also mechanisms utilized by this opportunistic pathogen to confer intrinsic drug resistance [64]. Classic antibiotic resistance can be conferred through mutations and/or horizontal gene transfer of antibiotic modifying enzyme genes, which can be acquired through plasmid transfer from one bacterial species to another [45]. Adaptive resistance, described as the transient resistance to antibiotics, typically emerges rapidly in the response to prolonged exposure to antibiotic treatment. Antibiotic resistance is a significant problem in clinical settings, and without novel treatment approaches, will only progress. A study determined that out of 60 clinical isolates of *P. aeruginosa* strains from burn patients, 90% were resistant to at least one antibiotic, while 94% were multidrug resistant [45].

Despite the urgent need for effective therapeutics, *P. aeruginosa* has been proven difficult to treat. A total of three vaccines against *P. aeruginosa* had been evaluated in Phase III clinical trials, none of which were deemed successful [65]. While numerous targets pertaining to *P. aeruginosa* pathogenesis have been explored, the high mutation and variability rate in targets such as LPS and flagellum present a major obstacle in vaccine design. The development of novel antibiotics and alternative therapeutics is extremely slow, in part due to the complexity of drug resistance, as well as certain roles in *P. aeruginosa* pathogenesis, which remain unclear [27]. This barrier, as well as the limited effective antibiotic treatments currently available, highlight the need for novel therapeutic approaches.

## 6. *P. aeruginosa* Specific mAbs

### 6.1. Murine Derived Antibodies

The first attempt at passive immunization for treatment against *P. aeruginosa* infections involved the infusion of anti-pseudomonas immune plasma to granulocytopenia dogs in 1976 [66]. Current developments and advances in antibody-based therapies have enabled the characterization of mAb targets against *P. aeruginosa* that can be adapted for human use. An extremely potent and clear candidate for pathogenic targeting is the PcrV protein, which is located at the tip of the T3SS and responsible for the transportation of cytotoxic effector proteins into the host cytoplasm [67]. Generated through immunization of purified PcrV in mice, mAb 166 is an IgG2bK antibody that binds to amino acids 158–217 of PcrV [68]. In an acute lung infection model, mAb 166 provided significant protection in animals challenged with up to 20-times the lethal dose of *P. aeruginosa* [68]. Administration of mAb 166 Fab fragments also resulted in a high level of protection (80% survival), further demonstrating the potency of this antibody [68].

### 6.2. Human Derived Antibodies

While murine anti-*P. aeruginosa* mAbs have shown promise in animal models, they are limited therapeutically due to their potential immunogenicity in humans [69]. Alternatively, highly potent neutralizing mAbs that have strong therapeutic and prophylactic efficacy against *P. aeruginosa* have been identified through high-throughput screening of B cells from convalescent patients. Lymphocytes from healthy volunteers and CF patients were transformed and cultured to generate lymphoblastoid cell lines secreting human monoclonal antibodies MAB-RM5, MAB-FDD7 and MAB-11F9 against LPS from *P. aeruginosa* [70]. These mAbs were shown to protect neutropenic mice against challenge with *P. aeruginosa* immunotypes 2, 4 and 5, whereas MAB-9H10 provided protection against immunotypes 3 and 6 [70]. The specificity of these mAbs suggests that they can recognize more than one carbohydrate determinant of the LPS or can recognize other determinants that are not serotype-specific [70]. Another human mAb targeting the LPS of *P. aeruginosa* serotype 5 was generated by cell fusion between human tonsillar lymphocytes and a mouse plasmacytoma cell P3-X63-Ag8-U1. This mAb, termed mAb P3D9, was shown to be highly protective in a mouse model experiment with a 50% protective dose (PD_50_) of 2.4 µg/mouse [71]. A study conducted by Pier et al., screened human mAbs isolated from a healthy patient immunized with a purified preparation of alginate from *P. aeruginosa* and identified mAb F429γ1 that was able to induce phagocyte dependent- killing and provide protection against lethal *P. aeruginosa* pneumonia challenge with both mucoid and nonmucoid strains in a murine model [72]. A study conducted by DiGiandomenico et al., screened a library of human mAbs isolated from healthy individuals and patients convalescing from *P. aeruginosa* infections for OPK and cell attachment interference and identified Cam-003 [73]. This human mAb proved efficacious in multiple murine models of *P. aeruginosa* infection, including pneumonia, thermal injury, and ocular keratitis. The anti-PcrV human mAb, V2L2MD, demonstrated good prophylactic protection in several models of *P. aeruginosa* infection as well as in a post infection therapeutic model [74]. Interestingly, V2L2MD mediated significantly (*p* < 0.0001) better in vivo protection than that provided by another potent anti-PcrV murine mAb, mAb166 [74].

### 6.3. Bispecific Monoclonal Antibody Therapy

The concept of a bispecific antibody (bsAb) was first described by Nisonoff and colleagues in the 1960s, although it was not until 1996 that platforms for the generation of bispecific antibodies were developed [75]. Bispecific antibodies are clinically superior to mAbs due to their possession of two binding sites, directed at two different antigens or epitopes on the same or different target antigen. Progression in antibody engineering has led to the development of five bispecific mAbs that have received FDA approval, with an additional ~180 types of bsAb in various stages of clinical development [76,77]. Currently approved bsAbs are involved in tumor immunotherapy (Blinatumomab [77], Amivantamab [78]) and the treatment of hemophilia A (Emicizumab) [79], macular degeneration and diabetic related macular oedema (Faricimab) [80]. While bispecific mAbs against viral targets have been reported, including SARS-CoV-2 [81], limited bispecific mAbs have been generated to target bacterial antigens [82]. MEDI3902 is a promising bispecific antibody against the bacterial pathogen *P. aeruginosa*, combining the single-chain variable fragments (scFvs) of both anti-Psl and anti-PcrV mAbs. MEDI3902 has been shown to be efficacious in the prevention and treatment of *P. aeruginosa* infections in vivo. Treatment with MEDI3902 has shown decreased bacterial burden and decreased expression of pro-inflammatory mediators in lung tissue of multiple animal models [75].

### 6.4. Clinical Testing of Anti-P. aeruginosa mAbs

To date, a total of six *P. aeruginosa* mAbs have been evaluated in clinical trials, with none currently approved for the market (Table 2). Past and present candidates include AR-105 (Aerucin), a broadly active hIgG1 mAb targeting *P. aeruginosa* alginate, KB001-A, a pegylated anti-PcrV antibody, and KB001, a humanized anti-PcrV Fab’ fragment that was derived from the mouse mAb166, discussed previously. While all these clinical candidates were determined to be safe and well tolerated in humans, none were clinically effective when compared to the placebo group [83,84,85]. AR-101 (Panobacumab) is an IgM mAb that binds the LPS of O11 serotype *P. aeruginosa* and is currently in clinical development as an adjunctive therapy for hospital-acquired pneumonia (HAP) [86]. An IgY avian polyclonal antibody has been shown to promote bacterial opsonization and phagocytosis through prophylactic oral treatment (gargling) [87]. Phase III clinical trial of this therapeutic antibody has been completed, and the primary endpoint was not met [88].

One bispecific hIgG antibody (MEDI3902) holds great promise due to its antigenic binding affinity to the potent PcrV protein, and the abundantly expressed Psl exopolysaccharide that is crucial for biofilm formation in *P. aeruginosa* infections [82]. MEDI3902 combines the single-chain variable fragment (scFv) of both anti-Psl (Cam-003) and anti-PcrV (V2L2MD) mAbs, and although both mAbs could theoretically be administered in tandem, it is more practical to develop a single molecule clinical candidate [82]. A geographical survey of 269 *P. aeruginosa* clinical isolates demonstrated that between 97.3–100% of isolates expressed either one or both targets [82], therefore suggesting that the bispecific mAb would provide broad protection against *P. aeruginosa* infections in a nosocomial setting. MEDI3902 was found to be safe in a Phase 1 dose escalation study in healthy subjects [89], and was evaluated in a Phase II (NCT02696902) clinical trial in subjects at risk for *P. aeruginosa* pneumonia [90]. A single dose of MEDI3902 administered intravenously was shown to provide efficacy trends in subjects with lower levels of baseline inflammatory biomarkers, suggesting that this treatment may be geared towards certain individuals possessing lower baseline inflammation [91]. Further testing is required to fully elucidate the potential of this bsmAb for the protection and prevention of *P. aeruginosa* infections.

**Table 2 biomedicines-10-03162-t002:** Top therapeutic *P. aeruginosa*-specific mAbs.

Name/Antibody Compound	Antigenic Target	Clinical Trial	Results	Reference
Panobacumab (AR-101, KBPA-101) Human IgM mAb	LPS (O11 serotype)	Phase 1/2 (NCT00851435): Safety and tolerability in patients with hospital acquired pneumonia caused by serotype O11 *P. aeruginosa*	Phase 2a completed. Panobacumab adjunctive immunotherapy may improve clinical outcome in a shorter time if patients receive the full treatment.	[83]
MEDI3902 Bispecific human IgG mAb	PcrV and Psl	Phase 1 (NCT02255760): Dose escalation study of MEDI3902 evaluating safety, pharmokinetics, antidrug antibody responses, ex vivo anticytotoxicity and opsonophagocytic killing activities in healthy adults Phase 2 (NCT02696902): Efficacy and safety of MEDI3902 in mechanically ventilated participants for the prevention of nosocomial *P. aeruginosa* pneumonia	Endpoint not met. A single IV dose of MEDI3902 did not achieve primary efficacy endpoint of reduction in PA pneumonia	[89,91]
Aerucin (AR-105) Human IgG mAb	Alginate (O11 serotpye)	Phase 1 (NCT02486770): Safety evaluation of Aerucin in healthy individuals Phase 2 (NCT03027609): Efficacy, safety, and PK evaluation of Aerucin in combination with antibiotic treatment in *P. aeruginosa* ventilator-associated pneumonia (VAP) patients	Endpoint not met. No difference noticed between Aerucin and placebo patient groups for treatment of *P. aeruginosa.* Single IV infusion of Aerucin VAP patients.	[85]
KB001-A Humanized pegylated Fab’ fragment	PcrV	Phase 1/2 (NCT01695343): Evaluation of the effect of KB001-A on time-to-need for antibiotic treatment of CF patients	Endpoint not met due to lack of clinical efficacy. KB001-A was not associated with an increased time to need for antibiotics	[92]
KB001 Humanized IgG Fab’ fragment	PcrV	Phase 1/2 (NCT00691587): Safety and pharmacokinetics (PK) of KB001 in mechanically ventilated ICU patients colonized with *P. aeruginosa*. Phase 1/2 (NCT00638365): Dose escalation study of KB001 in CF patients colonized with *P. aeruginosa*.	Endpoint not met. No signify can’t differences in the placebo group and the PA colonized CF patients.	[84]
PsAer-IgY Avian IgY pAb	Flagellin	(Phase 1/2) NCT00633191: Study of anti-pseudomonas IgY in prevention of recurrence of *P. aeruginosa* infections in CF Patients. Phase 3 (NCT01455675): Efficacy Study of IgY in Cystic Fibrosis Patients	Clinical efficacy results were unclear. IgY antibodies were present in the oral cavity for over 24 h after oral administration.	[88]

## 7. Prospects for AAV-Mediated Monoclonal Antibody Expression for Prevention of *P. aeruginosa* Infections

### 7.1. AAV Vectored Immunoprophylaxis (VIP)

AAV gene therapy vectors are currently the most promising in vivo gene delivery vehicles. AAV has gained significant traction as a gene therapy vector due to its lack of pathogenicity, low-level genomic integration, and favorable delivery and sustained transgene expressing properties [93]. To date, three AAV therapies have received regulatory approval, including Glybera, for the treatment of lipoprotein lipase deficiency, Luxturna, a treatment for vision loss attributed to inherited retinal dystrophy, and Zolgensma, a treatment for spinal muscular atrophy [94]. The genome of AAV can be easily manipulated to allow for in vivo expression of therapeutic transgenes for extended periods of time (>10 years) [95]. Recombinant AAV (rAAV) consists of the same capsid and genome structure found in wildtype AAV, with the primary difference being the removal of the AAV protein-coding sequences to make space for the promoter and therapeutic transgene [96]. The Rep and Cap proteins are instead supplied in *trans*, allowing for production of replication defective AAV particles.

AAV-vectored expression of highly potent, pathogen-specific mAbs is a promising method to enable long-term expression of therapeutic antibodies in vivo following intramuscular (IM) injection. The AAV genome is engineered to express the heavy and light chain domains of the mAb as one polyprotein separated by a 2A self-processing peptide sequence [97] that contains a furin cleavage site (F2A) at the amino-terminus for optimal processing [98,99]. This results in the rapid and robust expression of full length mAbs indistinguishable from those produced by the endogenous immune response [100,101]. AAV-mediated mAb expression has been shown to protect against a variety of infectious diseases including human immunodeficiency virus (HIV), malaria, respiratory syncytial virus, influenza, Ebola virus, and neonatal herpes simplex virus, in a range of animal models [100,102,103,104,105,106,107,108,109,110] and is now being tested in two human clinical trials for prevention of HIV (NCT03374202 and NCT01937455).

Although the initial application of VIP was directed towards prevention of HIV, this strategy has now been tested against an ever-expanding list of infectious diseases, including more recently, bacterial infections like *C. difficile* [111]. Based on the expanding list of successful pre-clinical applications of AAV-VIP, utilizing this platform as a viral vector for the delivery of pre-characterized anti-pseudomonal antibodies holds promise as a therapeutic treatment, which could be used as an adjunct therapy alongside antibiotic treatment.

### 7.2. Clinical Relevance of AAV VIP

Numerous advanced-stage clinical trials are currently ongoing to evaluate the efficacy of AAV-therapies for a variety of genetic disorders, including alpha-1 antitrypsin deficiency, choroideremia, and hemophilia. Along with these clinical trials are two AAV-mAb platforms that are being evaluated for safety and efficacy with respect to prevention or treatment of HIV infections. The first clinical trial (NCT01937455) evaluating AAV-mediated antibody gene delivery for the prevention of HIV infection utilized an AAV1 vector expressing the broadly neutralizing anti-HIV antibody (bnAb) PG9 [112], that has been shown to inhibit a panel of HIV-1 viruses at much lower doses than competing broadly neutralizing antibodies [113,114]. This first generation bicistronic vector was designed to express the heavy and light chain of PG9 from the CMV and EF1a promoter, respectively [115]. Patients enrolled in the study (HIV negative men 18–45 years of age) were administered between 4 × 10^12^ vg to 1.2 × 10^14^ vg of rAAV1-PG9DP intramuscularly. No serious adverse events were noted, with only a singular mild adverse event attributed to the scheduled treatment regimens. PG9 mRNA transcripts were detectable in muscle biopsies as was IgG expression within muscle cells and extracellular tissues via immunohistochemistry. However, serum concentrations of PG9 were below the limit of detection of the ELISA, which was 2.5 µg/mL. Anti-drug antibody (ADA) responses were detected in many of the participants, particularly in the high dose group, which likely contributed to the lack of PG9 in the serum [112]. Further optimization of the AAV capsid, genome cassette design, antibody selection, and dose may need to be revised to mitigate anti-drug immune responses in future trials.

A second Phase I AAV IP clinical trial (NCT03374202) evaluating AAV-mAb expression, conducted by David Baltimore and colleagues at the Ragon Institute of MIT and Harvard and Massachusetts General Hospital, evaluated the safety and efficacy of AAV8-mediated VRC07 expression in HIV-infected adults (ages 18–60) [116]. In this trial, the AAV genome was designed to express the heavy and light chain domains of VRC07 separated by an F2A self-cleaving peptide from a single composite CASI promoter [100,116]. Participants were administered between 5 × 10^10^ to 2.5 × 10^12^ vg/kg of AAV8-VRC07 via intramuscular injection. For this study, 50 μg/mL of VRC07 was chosen as the target serum concentration based on previous data in mice and NHPs [117], and the fact that a serum concentration of 50 μg/mL post-AAV8-VRC07 administration resulted in sustained serum concentrations of 3–5 μg/mL, which have previously been shown to decrease HIV viral loads in humans [118]. No adverse events were reported following administration of AAV8-VRC07, establishing the safety and tolerability of this AAV VIP therapy. Three out of eight participants reached maximum serum VRC07 concentrations of >1 µg/mL, with four sustaining maximal serum VRC07 concentrations three years after receiving AAV8-VRC07. ADA responses towards the Fab portion of the VRC07 were detected in three of the participants, and this correlated with a decrease in serum VRC07 levels in two of the individuals. Furthermore, anti-AAV8 capsid immune responses were reported in a dose dependent manner [116]. This is the first study demonstrating statistically significant, long-term sustained production of bnAbs in HIV-infected individuals; however, further development to decrease ADAs and anti-capsid immune responses will likely result in increased serum transgene expression.

### 7.3. Benefits of AAV VIP

AAV-mediated expression of mAbs either as a prophylactic or therapeutic provides an alternative to traditional vaccines or passive mAb therapies, respectively. Development of such AAV-mAb therapies can begin as soon as sequences for pathogen-specific neutralizing mAbs become available and may be an ideal stop gap measure for intervention in outbreaks where a vaccine is unavailable and could potentially be used in combination with a traditional vaccine to combine the benefits of both vaccination strategies. Through a singular administration, AAV-mAb potentially provides a longer-lasting therapeutic level of mAbs compared to passive administration. Since AAV-mAb expression leads to production of protective mAbs independent of the immune system, it could be utilized in people who do not respond to traditional vaccines or whose immune system is functioning poorly, including immunocompromised or elderly individuals The AAV capsid can be swapped or modified to target specific cell types and tissues, or to evade pre-existing immunity to the capsid [119]. Lastly, AAV can be stored at a variety of temperatures, while maintaining capsid integrity, and can be lyophilized to facilitate long-term storage [120]. AAV-mAb expression is an ideal stop-gap measure for intervention in outbreaks where a vaccine is unavailable (e.g., Marburg and Lassa virus) and could perhaps be used in combination with a traditional vaccine to combine the benefits of both vaccination strategies.

### 7.4. Caveats of AAV-VIP

While AAV VIP therapy holds great promise as an alternative or adjunctive therapeutic for *P. aeruginosa* infections, hurdles to this approach still exist. Prior exposure to wild-type AAV can lead to the development of both humoral and T cell immunity that may result in neutralization of the vector or immune-mediated clearance of vector transduced cells, which can affect the duration of transgene expression. De novo immune responses to the administered AAV vector represent another impediment to the success of AAV VIP [121]. TLR2 and TLR9 are activated through sensing of the AAV capsid and genomic DNA, respectively, which can decrease transgene expression and increase immunogenicity [122,123]. However, there are many methods that can be employed at the time of AAV administration to suppress immune responses to the vector [124,125]. For example, co-administration of AAV vectors with rapamycin containing nanoparticles has been shown to block the generation of both capsid- and transgene-specific T cell responses [126,127] and pre-treatment with IgG-cleaving endopeptidases enables in vivo transduction and repeat administration in the presence of anti-AAV neutralizing antibodies [128]. With AAV being an extremely relevant viral vector in gene therapy, extensive research is being conducted to develop immune response mitigating strategies that might provide solutions to this issue plaguing AAV gene therapies.

As previously mentioned, ADA responses have been documented in NHP studies and human clinical trials, which can be attributed to vector design and the immunogenicity of the transgene product itself [112]. The caveat to these studies is that all NHP studies and humans clinical trials evaluating AAV VIP have been done using HIV bNAbs, which have undergone extensive somatic hypermutation and possess long, variable heavy-chain third complementarity-determining regions [129] which are potentially more immunogenic than mAbs isolated from human survivors of acute viral infections, such as Ebola or Marburg virus [130]. Therefore, it is important to develop alternative models to evaluate AAV VIP so that ADA responses to mAbs can be characterized and methods to induce tolerance and mitigate the formation of ADA can be evaluated.

Finally, it may be necessary to implement genomic safety measures that can excise or invert sections of the AAV genome upon transient drug administration to turn off AAV-mAb expression should long-term mAb expression be undesirable [131,132,133]. It should also be noted that the cost of vector production is still a formidable barrier to bringing gene therapies to the clinic.

## 8. Conclusions

*P. aeruginosa* is a clinically relevant bacterial pathogen associated with nosocomial infections, high infection rates in immunocompromised individuals, and chronic infections in CF patients. The stealth virulence factors utilized by *P. aeruginosa,* in addition to increasing antimicrobial resistance, have resulted in a pathogen of public health concern lacking an effective vaccine.

Since the introduction AAV-VIP in the early 2000s [134], there has been an influx of vectored therapies extending to infections and applications beyond HIV [134]. Continued engineering of AAV vectors to increase mAb expression in vivo, through modification of various genome elements (such as promoters or poly adenylation signals or post-transcriptional factors), codon optimization, addition of host immune response dampening elements (such as inflammation-inhibiting oligonucleotide 2) or evolving capsid sequences for enhanced immune evasion [135], will aid in pathogen-specific advantages to the AAV-VIP platform.

The field of antibody engineering has resulted in an ever-increasing number of designs for improved therapeutic antibody expression. Multidomain antibodies, such as diabodies or bispecific mAbs, allow for variable domains of multiple potent antibodies to be structurally bound together, for binding and neutralizing of multiple epitopes from the same protein [136,137]. These multifaceted antibody proteins could result in an AAV-VIP therapy that could neutralize *P. aeruginosa* at multiple pathogenic epitopes from a single product.

We previously demonstrated AAV-mediated expression of *C. difficile* toxin-specific antibodies protects mice from toxin challenge demonstrating the potential of AAV VIP as an alternative or adjunctive treatment for recurring bacterial infections [111]. The coupling of potent *P. aeruginosa*-specific mAbs with AAV provides an additional tool in the currently limited arsenal of *P. aeruginosa* treatments. Here, we delve into the use of VIP for prevention and treatment of *P. aeruginosa*; however, this platform could be applied to other bacterial infections where effective treatments remain elusive. With the recent FDA approvals of Luxturna and Zolgensma, AAV gene therapies are rapidly becoming tools for mainstream medicine, which holds great promise for the field of VIP.

## Figures and Tables

**Table 1 biomedicines-10-03162-t001:** Conventional classes of antibiotics for *P. aeruginosa* infection.

Antibiotic Class	Antibiotic	Mechanism of Action	Reference
B-lactams	Penicillin Cephalosporins Carbapenems	Inhibit bacterial cell wall biosynthesis	[57]
Aminoglycosides	Tobramycin Gentamycin Amikacin Plazomicin	Inhibit bacterial protein synthesis	[1]
Quinolones	Ciproflaxin Levofloxacin Pefloxacin	Inhibit bacterial DNA replication	[55]
Polymyxins	Polymyxin B Colistin	Destroy LPS and disrupt outer cell membrane	[58]

## Data Availability

Not applicable.

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
