# Peer review of "Overcoming Barriers to Preventing and Treating P. aeruginosa Infections Using AAV Vectored Immunoprophylaxis"

_biomedicines, 2022, doi:10.3390/biomedicines10123162_

Round 1
Reviewer 1 Report
In the manuscript “Overcoming barriers to preventing and treating P. aeruginosa infections using AAV vectored immunoprophylaxis” the authors evaluated the prevalence and antibiotic resistance of P.aeruginosa. They analyzed the impact of P.aeruginosa-related infections on peopole, focusing their attention on patients with cystic fibrosis. The manuscript is well-structured and organized. it shows a detailed summary of issues related to infections caused by gram-negative bacteria, especially P.aeruginosa. It places increased attention on antibiotic resistance and the difficulty in treating bacterial infections, especially in frail patients.
Reviewer 2 Report
The manuscript entitled "Overcoming barriers to preventing and treating P. aeruginosa infections using AAV vectored immunoprophylaxis" is very significant as the mentioned bacteria is one of the dreaded pathogens. I have the following comments before the manuscript gets accepted for publication.
1. Authors have discussed ESKAPE pathogens to the significance of the topic. P. aeruginosa is also a significant pathogen of SPEAKS pathogen, which have reported recently. Authors should cite these two papers and discuss the SPEAKS pathogens briefly.
a. https://www.mdpi.com/2079-6382/11/7/939
Antibiotics 2022, 11(7), 939; https://doi.org/10.3390/antibiotics11070939
b. https://www.sciencedirect.com/science/article/pii/S0140673621027240
2. There are many typos and minor corrections. Please find the attached file.
